# Insight into Isomeric Diversity of Glycated Amino Acids in Maillard Reaction Mixtures

**DOI:** 10.3390/ijms23073430

**Published:** 2022-03-23

**Authors:** Haoran Xing, Varoujan Yaylayan

**Affiliations:** Department of Food Science & Agricultural Chemistry, McGill University, 21111 Lakeshore, Ste Anne de Bellevue, QC H9X 3V9, Canada; haoran.xing@mcgill.ca

**Keywords:** Schiff base, Amadori compounds, MS/MS diagnostic ions, monoglycated amino acid, diglycated amino acid, mechanochemistry, high-resolution mass spectrometry (HRMS)

## Abstract

Maillard reactions generate a wide array of amino acid- and sugar-derived intermediates; the isomeric mixtures of glycated amino acids are of particular interest. Excluding stereoisomers, regioisomers, and various anomers, most amino acids can form two monoglycated and three *N*,*N*-diglycated isomers when reacted with sugars during the Maillard reaction. Using synthetic Schiff bases and Amadori compounds as standards, we have demonstrated that diagnostic ions obtained from MS/MS fragmentations in negative ionization mode can be used effectively for the discrimination between glucose-derived Schiff bases and their corresponding Amadori compounds in both mono- and diglycated forms. The utilization of these diagnostic ions and isotopic labeling in the glycine/glucose model system revealed that milling glucose/glycine mixtures for 30 min/30 Hz at ambient temperature produced monoglycated glycine in equal proportions of Amadori and Schiff base forms, whereas diglycated glycine was a mixture of the three isomers: Schiff-Schiff, Schiff-Amadori, or Amadori-Amadori in approximately equal molar proportions. The above results were further corroborated using a synthetic histidine Amadori product, *N*,*N*-difructosyl-β-alanine, dipeptides, and ribose. Using mechanochemistry as a convenient synthetic tool in combination with MS/MS diagnostic ions, the isomeric diversity of the early stages of the Maillard reaction can be revealed.

## 1. Introduction

Maillard reaction mixtures can produce a wide array of compounds, many of which are isomeric in nature, particularly the glycated amino acids. Excluding stereoisomers, regioisomers, anomers, and numerous conformations and open-chain forms, most amino acids can generate two monoglycated and three *N*,*N*-diglycated isomers. The development of efficient analytical methodologies for their analysis remains a formidable task. However, high-resolution mass spectrometry (HRMS) is emerging as a suitable analytical platform to perform such analyses [1,2,3]. Furthermore, unequivocal structural elucidation can be achieved during subsequent tandem mass spectrometry (MS/MS) through collision-induced disassociation (CID) and isotope labeling techniques. However, there is limited information that can be found in the literature on the utilization of MS/MS fragmentations for the purpose of distinguishing isomeric intermediates formed in the Maillard reaction [4,5,6,7]. Knowledge of the relative amounts of glycated amino acid isomers in these reaction mixtures can be a useful strategy for estimating their relative importance. The ability to discriminate between initially formed isomeric intermediates in the Maillard reaction mixtures using MS/MS can provide researchers in the field of “omics” with an important tool for more in-depth analysis of such reactions. The success of this approach relies heavily on the availability of synthetic standards. A recent approach utilizing mechanochemistry as a synthetic tool to access reactive and hard to obtain intermediates such as Schiff bases and Amadori compounds formed during the Maillard reaction was proposed [8].

Mechanochemistry has recently emerged as a popular method for conducting various organic transformations in the solid state without bulk solvation and thermal energy input [9]. Not only has this technique been recognized as “greener” and more energy-efficient compared to solution-based thermochemistry, it can also provide opportunities to synthesize hard-to-obtain compounds [10]. Unlike the thermally catalyzed solvent phase Maillard reaction, where the important intermediates generated are usually degraded or polymerized at the end of the reaction with little or no opportunity to be isolated or even identified, reactions performed under ball-milling conditions are more likely to retain many of the reactive intermediates due to the shorter timescale of the reaction and the near-room temperature conditions. Furthermore, the reaction generates higher yields due to solvent-free conditions and the lack of a solvent interference. Mechanochemistry by ball milling is therefore proposed as a promising tool to generate standards or their mixtures needed to study the isomeric complexity of the glycated amino acids in the Maillard reaction [8,11]. Taking advantage of the ability of the side chain of different amino acids to direct the glycation towards the formation of either Schiff bases or Amadori rearrangement products, we utilized mechanochemistry as a synthetic tool in conjunction with independently synthesized standards to generate a series of glycated amino acid mixtures and investigate the utility of MS/MS diagnostic ions in distinguishing these isomeric intermediates (Schiff bases and Amadori rearrangement products) under different MS/MS fragmentation conditions [8,12]. 

## 2. Results

A general summary of the diagnostic ions (indicated in the square boxes) that were generated from the MS/MS fragmentation of monoglycated and diglycated amino acids under negative-mode ESI is shown in Figure 1. These diagnostic ions were successfully employed to distinguish Schiff bases and their Amadori isomers in selected mono- and diglycated amino acids [8,12,13,14], such as glycine, phenylalanine, isoleucine, histidine, arginine, glutamic acid, and aspartic acid [8,12], and similarly in the glycated N_ε_-moiety of lysine [13], and dipeptides (i.e., glycylglycine and carnosine).

### 2.1. Convenient Generation of Mono- and Diglycated Amino Acids Using a Mechanochemical Approach

For a reaction time as short as 30 min at 30 Hz, the mechanochemical reactions induced by ball milling lead to the formation of a significant amount of monoglycated glycine observed at *m*/*z* 236, as well as the diglycated glycine observed at *m*/*z* 398 with considerable intensity (see Figure 2a). For the hydrothermal reaction in water (50% *w*/*v*) at 120 °C for 1 h, the reaction mixture consisted mainly of unreacted sugar, trace amounts of monoglycated glycine, and no indication of any diglycated adduct formation (see Figure 2b). Generally, the overall profiles of mechanochemical Maillard reaction mixtures are simple, consisting mainly of free sugar, free amino acid, mono-, and diglycated condensation adducts, and minor degradation products (see Figure 2 and Figure 3). This observation is true not only for the majority of α-amino acids studied (glycine, phenylalanine, isoleucine, histidine, arginine, glutamic acid, and aspartic acid) [8], but also for dipeptides such as glycylglycine and carnosine (Figure 3b,c) and for reactive amino acids [13] and sugars such as lysine and ribose (Figure 3a).

### 2.2. Diagnostic MS/MS Fragmentation Patterns for Distinguishing Schiff Base and Amadori Isomers of Mono- and Diglycated Amino Acids

The diagnostic ions shown in Figure 1 are based on the detailed MS/MS analysis of a variety of synthetic standards, including glycine Schiff base [8], glycine ARP [8], proline Schiff base [8], proline ARP [8], alanine Schiff base, [8] *N_α_*-formyl lysine Schiff base [13], *N_α_*-formyl lysine ARP [13], *N*,*N*-difructosyl glycine (ARP-ARP), [14] *N_ε_*,*N_ε_*-difructosyl-N*_α_*-formyl-lysine (ARP-ARP) [13], and *N_α_*,*N_ε_*-difructosyl lysine (ARP-ARP) [13]. Here, we include diagnostic ions (see Figure 4) originating from synthetic histidine ARP and *N*,*N*-difructosyl-β-alanine (ARP-ARP). As shown in Figure 4a, the fragmentation of histidine ARP under negative ionization mode generated two major fragments; both are considered as Amadori diagnostic ions. The ion at *m*/*z* 226 is generated through retro-aldolization resulting in C3-C4 sugar chain cleavage, and the ion at *m*/*z* 154 is generated through β-elimination [8,13]. As expected, the predicted diagnostic ion for the histidine Schiff base at *m*/*z* 196 was absent, which was in agreement with our previous study [11] (Figure 4a). Similarly, a consistent fragmentation pattern was also observed in the case of synthetic *N*,*N*-difructosyl-β-alanine (ARP-ARP). Under MS/MS fragmentation, the molecular ion at *m*/*z* 412 generated four major fragment ions at *m*/*z* 322, 250, 232, and 160 (Figure 5). Interestingly, trace amounts of diagnostic ions for the Schiff-Schiff isomer (*m*/*z* 172) and for the Schiff-ARP isomer (*m*/*z* 202) were also observed. These ions were assumed to be formed through mutarotation of sugar moieties when the sample was dissolved in water-methanol during the ESI-MS analysis. 

### 2.3. Application of Diagnostic MS/MS Fragmentations for the Identification of Glycated Amino Acid Isomers

To demonstrate how the diagnostic ions proposed in Figure 1 can be utilized for the analysis of the isomeric diversity of ball-milled Maillard reaction mixtures, such as glycine/ribose, glycylglycine/glucose, and carnosine/glucose, the molecular ions corresponding to the glycated adducts were targeted for MS/MS fragmentation in negative ionization mode; the results are shown in Figure 6. According to this figure, monoribosylated glycine was primarily in the Amadori form, monoglycated glycylglycine was also mainly in the Amadori form with a small amount in the Schiff base form, and monoglycated carnosine was a mixture consisting more of the Schiff base than the Amadori form. The use of the same approach for studying the isomeric composition of diglycated lysine and diglycated glycine has been reported elsewhere [13,14]. Furthermore, the utilization of the ratios of the peak intensities of the diagnostic ions as a true indicator for the relative amount of Schiff bases to ARPs has also been confirmed [8].

## 3. Discussion

By conveniently accessing mixtures rich in Schiff base or Amadori compounds through mechanochemistry, we were able to identify characteristic MS/MS fragmentation patterns associated with Schiff bases or Amadori compounds [8,12]. Furthermore, through the milling of glucose with various amino acids, it has been concluded that amino acids with basic side chains generate mixtures rich in Schiff bases and amino acids with acidic side chains generate mixtures rich in Amadori compounds, while amino acids with neutral side chains generate both compounds in comparable proportions [8]. For example, aspartic acid and glutamic acid generated more Amadori products, whereas histidine and arginine generated more Schiff bases, and the neutral amino acids, such as glycine, isoleucine, and phenylalanine, generated both isomers equally [8]. In this study, we further demonstrate that the sugar type also dictates the type of isomer formed, i.e., the glycine/ribose model generates mainly ARPs (Figure 6a) whereas glycine/glucose generated equal amounts of each [8]. 

Although the tested amino acid/glucose mixtures selectively generated Schiff base or Amadori compounds in addition to a very low intensity of degradation products, some model systems also generated ions consistent with the molecular masses of diglycated adducts, further enhancing the synthetic reach of mechanochemistry. The formation of the diglycated amino acids is not surprising in the glucose/glycine model system due to the minimum steric hindrance of the glycine side chain [14] (Figure 2). Literature reports on the diglycated amino acids are scarce [14,15,16,17,18,19,20] due to their reactivity and transient nature. Factors including elevated temperature and the presence of solvents, which are necessary for initiating and maintaining the hydrothermal reactions, could play a part in destabilizing the elusive diglycated amino acids [20]. Mechanochemistry by ball milling made it possible to perform solvent-free chemical reactions without thermal energy input, thus preventing the thermal degradation and solvolysis of the diglycated amino acids. 

By combining high-resolution mass spectrometry (HRMS) and MS/MS analysis, we have demonstrated the isomeric diversity of the Maillard reaction mixtures generated by ball milling. As shown in Figure 1, all amino acids can generate two monoglycated (Amadori or Schiff) and three diglycated (Schiff-Schiff, Amadori-Amadori, and Amadori-Schiff) isomers. Using mass spectrometric analysis, one could readily distinguish monoglycated from diglycated amino acid derivatives through the observation of the expected mass shift of 162 Da per attached glucose moiety, proving that their structures could be verified through MS/MS analysis. Such diglycated ions were observed in all model systems generated by ball milling, but more so in glycine- and lysine-containing mixtures. Simple *N*,*N*-diglycated α-amino acids, such as glycine, could assume three possible isomeric configurations of Schiff-Schiff, Amadori-Amadori, and Amadori-Schiff, whereas amino acids with multiple amino groups, such as lysine, could generate monoglycated *N_α_*- and *N_ε_*-regioisomers and *N_α_*-, *N_ε_*-, or *N*,*N*-diglycated isomers of Schiff bases and of Amadori compounds. Although monitoring the mass shifts of 162 Da can provide some evidence for the formation of diglycated amino acids, further MS/MS analysis is required for the detailed characterization of the isomeric mixture. The diagnostic ions of Schiff bases and Amadori compounds shown in Figure 1 were identified using synthetic *N*,*N*-difructosyl glycine (ARP-ARP) [14], *N_ε_*,*N_ε_*-difructosyl-N*_α_*-formyl-lysine (ARP-ARP) [13], *N_α_*,*N_ε_*-difructosyl lysine (ARP-ARP) [13], and *N*,*N*-difructosyl-β-alanine (Figure 4b and Figure 5). Furthermore, we have confirmed their ability to characterize the isomeric composition of synthetic diglycated amino acids generated through ball milling. For example, using the diagnostic MS/MS fragmentations shown in Figure 1, we were able to distinguish between the Schiff-Schiff, Amadori-Amadori, and Amadori-Schiff isomers of glycine. Milling glucose/aglycine (1:1) for 30 min at ambient temperature produced diglycated glycine as a mixture of the three indicated isomers in approximately equal molar proportions. Lastly, milling of synthetic Schiff base or ARP of glycine with [U-^13^C]-glucose generated the same three diglycated isomers, with one isotopically enriched carbohydrate moiety in each [14]. Furthermore, ball milling of lysine and glucose selectively generated a mixture of mono-glycated adduct, mainly *N*_ε_-Schiff base and *N_α_*,*N_ε_*-diglycated adducts, predominantly as a mixture of Schiff-Schiff and Schiff-Amadori isomers. The dipeptides glycylglycine or carnosine behaved similarly when reacted with glucose (Figure 6), indicating the generality of the application of the diagnostic ions shown in Figure 1

## 4. Materials and Methods

### 4.1. Materials

Glycine (98%), glycylglycine (99%), carnosine (99%), d-glucose (99.5%), and d-ribose (99%) were purchased from Sigma-Aldrich Chemical Co. (Oakville, ON, Canada). N-(1-Deoxy-1-fructosyl)histidine (histidine ARP) was obtained from Toronto Research Chemicals Inc. (Toronto, ON, Canada). Synthesis and characterization of *N*,*N*-difructosyl-β-alanine was performed according to published procedures [18,21,22]. All other chemicals and reagents were of analytical grade from Fisher Scientific. Ultrapure water was used throughout the study. All materials were used without further purification.

### 4.2. Sample Preparation

d-Glucose or d-ribose and an amino acid or dipeptide mixtures were prepared at the 1:1 molar ratio. **Mechanochemical** reaction samples were prepared through ball milling (30 mg powder) at ambient temperature. All mechanochemical reactions were conducted in stainless steel grinding jars (10 mL) with 2 steel balls (3.2 mm in diameter; ball/sample ratio ~ 32:1) for creating inner friction. The jars were seated in the Retsch Mixer Mill (MM 400, Newtown, PA, USA) that performs radial oscillations in a horizontal position without coolant (the external jar temperature was  ~25 °C before milling and ~30 °C after milling) at a frequency of 30 Hz for 30 min. Samples collected after the reaction were stored at −20 °C for further analysis. **Hydrothermal reactions** were performed by heating the mixture in water (50% *w*/*v*) at 120 °C for 1 h in sealed stainless steel reactors.

### 4.3. Electrospray Ionization/Quadrupole Time of Flight/Mass Spectrometry (ESI/QqTOF/MS)

The diluted sample solutions (1 μL; 0.1 mg/mL) in 1:9 (*v*/*v*) water/methanol were supplied to the source directly via a syringe. The analysis was performed on a Bruker Maxis Impact quadrupole time-of-flight mass spectrometer (Bruker Daltonics, Bremen, Germany) operated in negative ion mode. Instrument calibration was performed using sodium formate clusters. The electrospray interphase settings were as follows: nebulizer pressure, 0.6 bar; drying gas, 4 L/min; temperature, 180 °C; and capillary voltage, 4500 V. The scan range was from *m*/*z* 50 to 1000. Tandem mass spectrometry (MS/MS) for the ions [M − H]^−^ under negative mode was carried out in MRM mode using collision energy ranging from 5 to 20 eV. The data were analyzed using Bruker Compass Data Analysis software (version 4.2) Bruker, Bremen, Germany. Each sample was analyzed in duplicate from independent trials.

## 5. Conclusions

The ratio of Amadori products to Schiff bases formed under specific reaction conditions is very much dependent on the moisture content, temperature, time, pH, and solvent; as such; it is very difficult to estimate this ratio, especially under hydrothermal reaction conditions where Schiff bases are quickly converted into Amadori compounds or hydrolyzed and Amadori compounds, in turn, are quickly degraded into various products. However, regardless of the reaction conditions, the initial or the nascent ratio of Schiff bases to Amadori compounds could perhaps be estimated, for a specific sugar/amino acid interacting pair, from the analysis of their corresponding solvent-free ball-milling glycated adducts. These conditions provide a temperature-independent pathway for the formation of the initial Schiff base and allow its conversion into Amadori product depending solely on the intrinsic acid/base properties of the particular amino acid side chain and the sugar moiety attached. 

To extract detailed information from the molecular ion representing isomeric glycated structures such as Schiff bases and Amadori compounds, characteristic MS/MS fragmentation patterns for each isomer are needed. The studies utilizing such proposed diagnostic ions have shown that they can be applied to characterize the isomeric composition of mono- as well as diglycated amino acids; for example, monoglycated glycine generated through ball milling was shown to exist in equal proportions of Schiff and Amadori forms, whereas the diglycated glycine was a mixture of Schiff-Schiff, ARP-Schiff, or ARP-ARP isomers in approximately equal abundance [14]. It could be proposed that under hydrothermal reaction conditions, the same proportion of glycated isomers is formed initially at the very beginning of the reaction from the same amino acids and sugars as under ball-milling conditions, but quickly dissipates into the more stable Amadori form or its degradation products depending on the time/temperature exposure. The extent of this conversion or decomposition of course depends on the specific reaction conditions. The knowledge of the initial composition of Schiff bases and Amadori products may help predict the chemical profile of their subsequent degradations and reactions. For example, the high content of Schiff bases relative to Amadori products may lead to the formation of more nitrogen-containing heterocyclic compounds compared to oxygen-containing heterocycles. If confirmed, such an approach can provide us with an insight into the intrinsic composition of the nascent glycated amino acids in hard-to-measure and rapidly changing environments such as the surfaces of meat or bread during grilling and baking, or even in biological systems. 

## Figures and Tables

**Figure 1 ijms-23-03430-f001:**
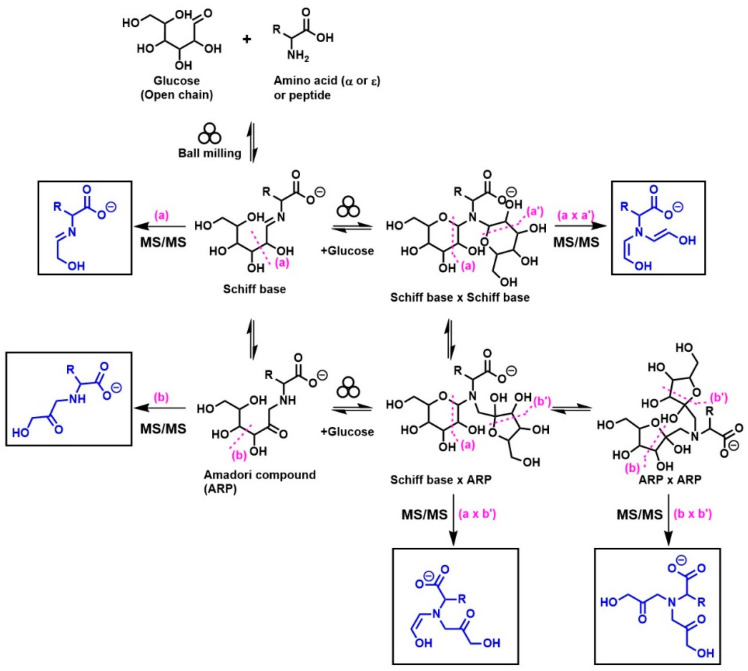
Structures of the MS/MS diagnostic ions (shown in square boxes) generated under negative-mode ESI that have been utilized for the identification of different isomers of monoglycated and diglycated amino acids. The specific tautomeric forms shown in the figure are for clarity only.

**Figure 2 ijms-23-03430-f002:**
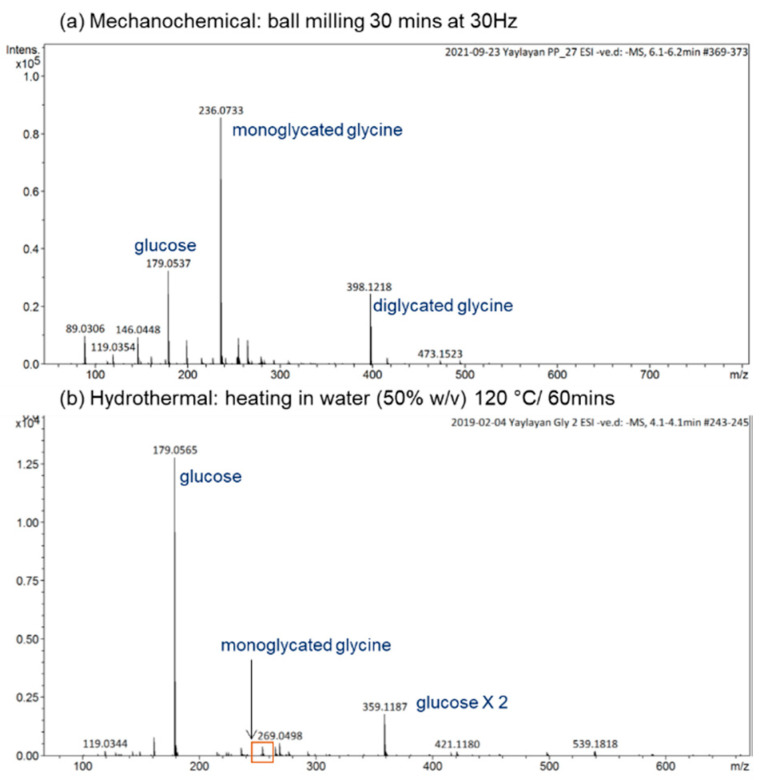
Comparison of the ESI(−ve)/MS spectra of the reaction mixtures of glycine/glucose (1:1) generated under (**a**) mechanochemical or (**b**) hydrothermal conditions. All identified ions are deprotonated [M − H]^−^.

**Figure 3 ijms-23-03430-f003:**
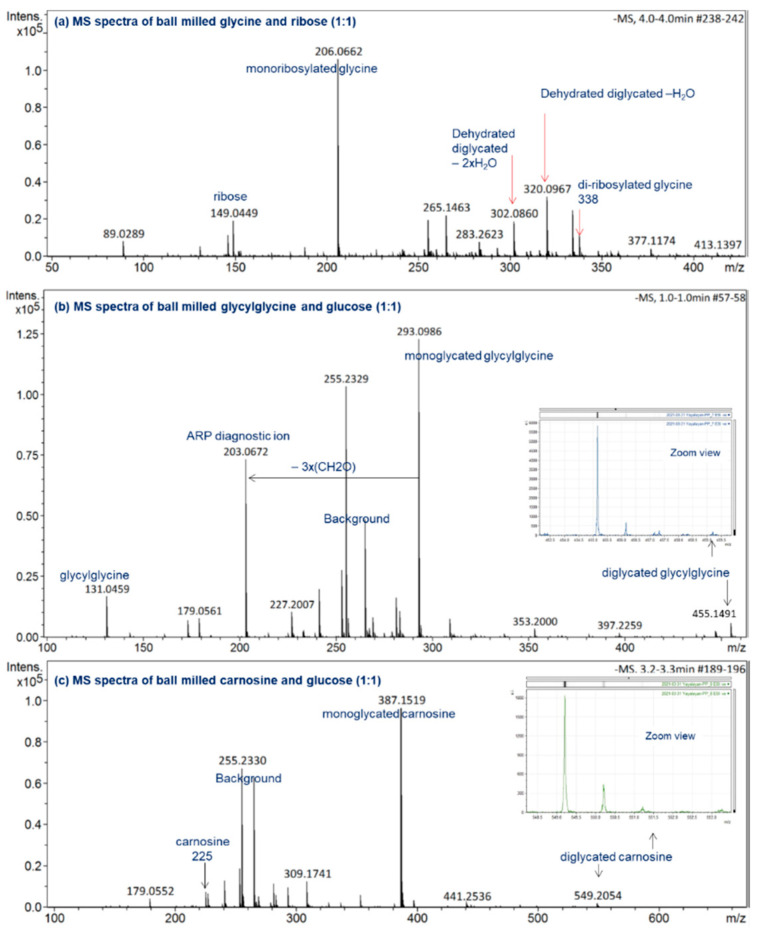
ESI(−ve)/MS spectra of ball-milled (**a**) ribose-glycine, (**b**) glucose-glycylglycine, (**c**) glucose-carnosine. All reaction mixtures were prepared at a 1:1 molar ratio and ball milled for 30 min at 30 Hz. All identified ions are deprotonated [M − H]^−^.

**Figure 4 ijms-23-03430-f004:**
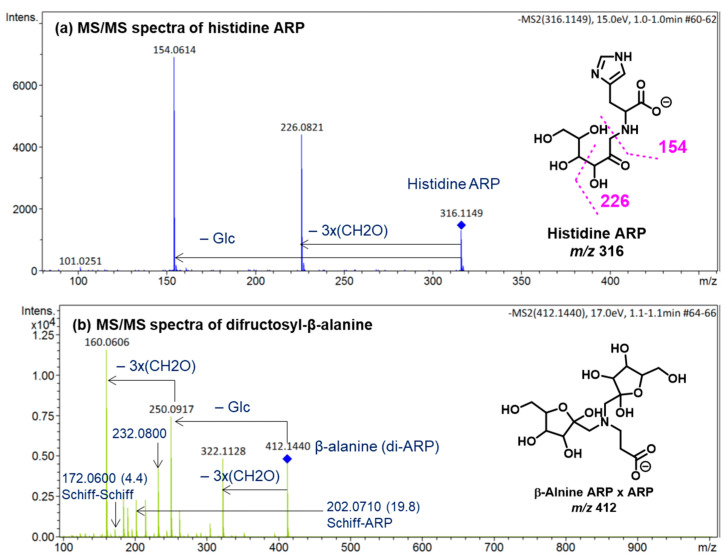
ESI (−ve)/MSMS spectra of molecular ions of the synthetic (**a**) histidine ARP and (**b**) *N*,*N*-difructosyl-β-alanine (ARP-ARP). All identified ions are deprotonated [M − H]^−^. The specific tautomeric forms shown in the figure are for clarity only.

**Figure 5 ijms-23-03430-f005:**
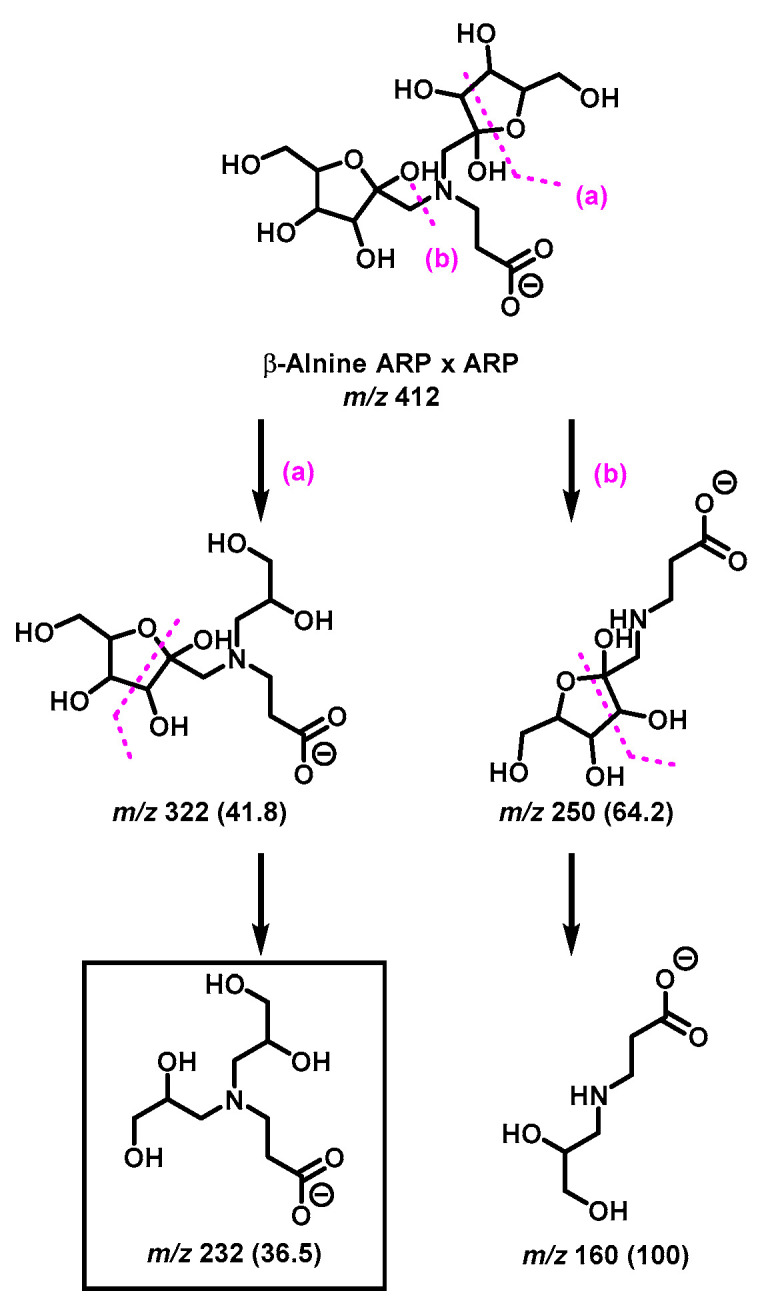
Proposed MS/MS fragmentation pathway of *N*,*N*-difructosyl-β-alanine (ARP-ARP) under negative ionization mode. Diagnostic ion for ARP-ARP is indicated in the square box. Relative abundances are shown inside the brackets.

**Figure 6 ijms-23-03430-f006:**
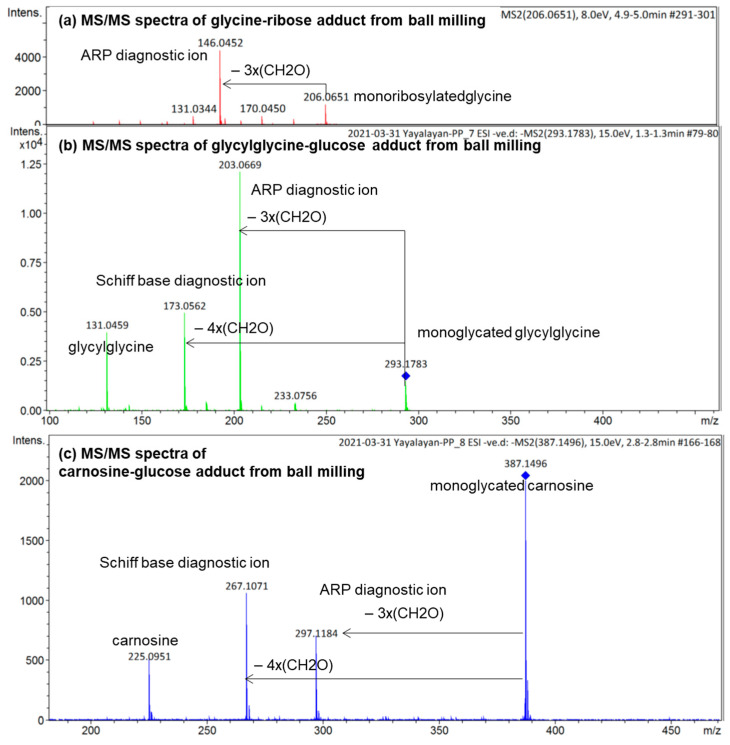
ESI (−ve)/MSMS spectra of the molecular ion of (**a**) monoribosylated glycine formed in ball-milled glycine/ribose; (**b**) monoglycated glycylglycine formed in ball-milled glycylglycine/glucose; (**c**) monoglycated carnosine formed in ball-milled carnosine/glucose. See Figure 3 for the corresponding MS spectra. All identified ions are deprotonated [M − H]^−^.

## Data Availability

Not applicable.

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
