# Peer review of "Insight into Isomeric Diversity of Glycated Amino Acids in Maillard Reaction Mixtures"

_ijms, 2022, doi:10.3390/ijms23073430_

Round 1

Reviewer 1 Report

The presented article is devoted to the MS/MS identification of the isomers of Maillard reaction. Since the Maillard reaction occurs during storage and thermal processing of food substances (amino acids in the presence of sugars), the study of this reaction is important from a practical point of view. The study of the distribution of isomers in the initial period of the reaction is a detailed study of the features of the reaction, which are mainly of theoretical interest. In this work, the Maillard reaction was carried out not in an aqueous medium with significant heating, but in dry form in mechanochemical conditions (ball milling) at room temperature. Without any doubt, the article will be of interest for the readers of IJMS.

I think that the article can be improved taking into account the following comments and suggestions:

- The use of tandem mass-spectrometry to identify isomers is well illustrated in the article. What are the prospects for using this method for the quantitative evaluation of isomers based on ion intensities, or for quantitative analysis, one should also use the chromatographic separation of isomers?

- How to control the moisture content of the powders that were used for the reaction, or whether this small amount of moisture does not matter?

- Difficult-to-measure media are mentioned, such as the surface of meat or bread during frying and baking (line 241-244).  It is not clear how the difficulty of taking samples from any object is related to the distribution of Schiff bases and Amadori compounds in early-stage of Maillard reaction. Since there is no conclusion chapter in the article, it is necessary to formulate in the final phase of the article exactly how the new analytical method will contribute to the study of the Maillard reaction.  

Reviewer 2 Report

Congratulation to the authors for you did a great work to obtain a method for detecting intermediate product of Mailard reaction, and I believe this should be helpful for understanding the Mailard reaction and advanced glycation endproducts. Please see attachment for more details

Reviewer 3 Report

Dear Sir or Madam,

the manuscript “Insight into isomeric diversity of glycated amino acids in the
Maillard reaction mixtures” describes a tandem mass spectrometric strategy to distinguish Schiff base and Amadori intermediates of amino acid glycation. The manuscript is strongly recommended for publication after considering just several remarks. The most important issue – the chemistry seems to be really different from that described in solution for peptides. Some more intensive discussion in the context of the published data is desired. And probably introduction can be extended in this sense.

  1. Just a general question: I have relatively low experience of MS/MS of amino-acid derived early glycation products, but quite rich experience with peptides. I never found double glycation at the side chain of lysine, as you show here. Can it be somehow related to mechanochemistry? How is relevant this for in vivo conditions? Could you comment on this?
  2. Figure 1. You show Amadori in the furanose form. My experience (also for side chains of lysine in peptides) tells me about betta-pyranose as the major form (according proton NMR). So, what is the basis of your decision for the furanose form?
  3. Line 249: why glucose is capitalized?
  4. Line 269: sorry for saying this, but quadrupole-time of flight is abbreviated as QqTOF – this mass analyzer has one filtering (Q1) and one RF-only (q2) quadrupoles. I realize, that many people don’t care now, but I thing it would be better to stick to good original style indroduced by the inventor of the QqTOF technology
  5. Line 273: how many negative modes has the instrument – please, check typos in the manuscript.
  6. m/z – italic everywhere please
  7. Please, make an abbreviation list.
  8. Figure 2. The peak labeling does not look good: please, label as [M-H]- and text label (both), but not in the way done now.
  9. Figure 4: why in one panel ARP is in the open form, whereas in the other – in the furanose one. Is it based on some NMR data (the panels are not labeled by the way)?
  10. It is very strange that the MS/MS spectra do not show conventional water and formaldehyde losses (see doi: 10.1002/jms.1117). Could you comment on this?
